# MESH-FREE EULERIAN PHYSICS-INFORMED NEURAL NETWORKS

## ABSTRACT

Physics-informed Neural Networks (PINNs) have recently emerged as a principled way to include prior physical knowledge in form of partial differential equations (PDEs) into neural networks. Although PINNs are generally viewed as mesh-free, current approaches still rely on collocation points within a bounded region, even in settings with spatially sparse signals. Furthermore, if the boundaries are not known, the selection of such a region is difficult and often results in a large proportion of collocation points being selected in areas of low relevance. To resolve this severe drawback of current methods, we present a mesh-free and adaptive approach termed particle-density PINN (pdPINN), which is inspired by the microscopic viewpoint of fluid dynamics. The method is based on the Eulerian formulation and, different from classical mesh-free method, does not require the introduction of Lagrangian updates. We propose to sample directly from the distribution over the particle positions, eliminating the need to introduce boundaries while adaptively focusing on the most relevant regions. This is achieved by interpreting a non-negative physical quantity (such as the density or temperature) as an unnormalized probability distribution from which we sample with dynamic Monte Carlo methods. The proposed method leads to higher sample efficiency and improved performance of PINNs. These advantages are demonstrated on various experiments based on the continuity equations, Fokker-Planck equations, and the heat equation.

## 1 INTRODUCTION

Many phenomena in physics are commonly described by partial differential equations (PDEs) which give rise to complex dynamical systems but often lack tractable analytical solutions. Important examples can be found for instance in fluid dynamics with typical applications in the design of gas and steam turbines (Oosthuizen & Carscallen, 2013), as well as modeling the collective motion of self-driven particles (Marchetti et al., 2013) such as flocks of birds or bacteria colonies (Szabó et al., 2006; Nussbaumer et al., 2021). Despite the relevant progress in establishing numerical PDE solvers, such as finite element and finite volume methods, the seamless incorporation of data remains an open problem (Freitag, 2020). To fill this gap, Physics-informed Neural Networks (PINNs) have emerged as an attractive alternative to classical methods for data-based forward and inverse solving of PDEs.

The general idea of PINNs is to use the expressive power of modern neural architectures for solving partial differential equations (PDEs) in a data-driven way by minimizing a PDE-based loss, cf. Raissi et al. (2019). Consider parameterized PDEs of the general form

$$f(t, \boldsymbol{x}|\boldsymbol{\lambda}) := \partial_t \boldsymbol{u}(t, \boldsymbol{x}) + P(\boldsymbol{u}|\boldsymbol{\lambda}) = 0, \tag{1}$$

where $P$ is a non-linear operator parameterized by $\boldsymbol{\lambda}$, and $\partial_t$ is the partial time derivative w.r.t. $t \in [0, T]$. The position $\boldsymbol{x} \in \Omega$ is defined on a spatial domain $\Omega \subseteq \mathbb{R}^d$. The PDE is subject to initial condition $g_0$

$$\boldsymbol{u}(0, \boldsymbol{x}) = g_0(\boldsymbol{x}) \tag{2}$$

for $\boldsymbol{x} \in \Omega$, and boundary conditions $g_{\partial\Omega}$

$$\boldsymbol{u}(t, \boldsymbol{x}) = g_{\partial\Omega}(\boldsymbol{x}) \tag{3}$$

for $\boldsymbol{x} \in \partial\Omega$ and $t \in [0, T]$. The main idea of PINNs consists in approximating $\boldsymbol{u}(t, \boldsymbol{x})$ (and hence $f(t, \boldsymbol{x})$) with a neural network given a small set of $N$ noisy observations $\boldsymbol{u}_{\text{obs}}$

$$\boldsymbol{u}(t^{(i)}, \boldsymbol{x}^{(i)}) + \epsilon^{(i)} = \boldsymbol{u}_{\text{obs}}^{(i)} \tag{4}$$

with noise $\epsilon^{(i)} \ll \boldsymbol{u}^{(i)} \ \forall i \in \{0, 1, \ldots, N\}$. This allows us to consider the following two important problem settings: If $\boldsymbol{\lambda}$ is known, the PDE is fully specified, and we aim to find a solution $\boldsymbol{u}$ in a data-driven manner by training a neural network. The PDE takes the role of a regularizer, where the particular physical laws provide our prior information. A second setting considers the inverse learning of the parameters $\lambda$ by including them into the optimization process in order to infer physical properties such as the viscosity coefficient of a fluid (Jagtap et al., 2020). Initial work on solving time-independent PDEs with neural networks with such PDE-based penalties was pioneered by Dissanayake & Phan-Thien (1994) and van Milligen et al. (1995), with later adoptions such as Parisi et al. (2003) extending it to non-steady and time-dependent settings.

**Loss functions.**    Typically, PINNs approximate $f(t, \boldsymbol{x})$ by the network $f_\Theta(t, \boldsymbol{x})$ in which the parameters $\Theta$ are adjusted by minimizing the combined loss of (i) reconstructing available observations ($L_{\text{obs}}$), (ii) softly enforcing the PDE constraints on the domain ($L_f$), and (iii) fulfilling the boundary ($L_b$) and initial conditions ($L_{\text{init}}$), i.e.

$$\Theta = \arg\min_\Theta \left[ w_1 L_{\text{obs}}(\boldsymbol{X}, \boldsymbol{t}, \boldsymbol{u}_{\text{obs}}, \Theta) + w_2 L_f(\Theta) + w_3 L_b(\Theta) + w_4 L_{\text{init}}(\Theta) \right], \qquad (5)$$

with loss weights $w_i \in \mathbb{R}_{\geq 0}$. A common choice for $L_{\text{obs}}$, $L_b$, and $L_{\text{init}}$ is the expected $\mathrm{L}^2$ loss, approximated via the average $\mathrm{L}^2$ loss over the observations and via sampled boundary and initial conditions, respectively. It should be noted that the formulation of the forward and inverse problem are identical in this setting, as observations and initial conditions are implemented in a similar manner.

**Enforcing the PDE.**    Although PINNs are by nature mesh-free, the PDE loss $L_f$ in Eq. 5 used for the soft enforcement of Eq. 1 requires a similar discretization step for approximating an integral over the continuous signal domain,

$$L_f(\Theta) = \frac{1}{|[0,T] \times \Omega|} \int_{t=0}^{T} \int_\Omega \|f_\Theta(t, \boldsymbol{x})\|_2^2 \, d\boldsymbol{x} \, dt = E_{p(t,\boldsymbol{x})} \left[ \|f_\Theta(t, \boldsymbol{x})\|_2^2 \right] \approx \frac{1}{n} \sum_{i=1}^{n} \|f_\Theta(t_i, \boldsymbol{x}_i)\|_2^2 \quad (6)$$

with $p(t, \boldsymbol{x})$ being supported on $[0, T] \times \Omega$. The points $\{(t^{(j)}, \boldsymbol{x}^{(j)})\}_{j=1}^n \subset [0, T] \times \Omega$ on which the PDE loss is evaluated are commonly referred to as *collocation points*. This formulation of PINNs for solving Eq. 1 is an Eulerian one, as the function $f_\Theta$ is updated by evaluating the PDE with respect to collocation points fixed in space. Initial approaches for selecting the collocation points in PINNs relied on a fixed grid (Lagaris et al., 1998; Rudd, 2013; Lagaris et al., 2000), followed up by work proposing stochastic estimates of the integral via (Quasi-) Monte Carlo methods (Sirignano & Spiliopoulos, 2018; Lu et al., 2021; Chen et al., 2019) or Latin Hypercube sampling (Raissi et al., 2019). However, these approaches to Eulerian PINNs cannot be directly applied if there are no known boundaries or boundary conditions, e.g. for $\Omega = \mathbb{R}^d$. Additionally, problems can arise if the constrained region is large compared to the area of interest. Considering for example the shock wave (of a compressible gas) in a comparably large space, most collocation points would fall into areas of low density. We argue that due to the locality of particle interactions, the regions with higher density are more relevant for regularizing the network.

To address these shortcomings of previous methods, we propose a mesh-free and adaptive approach for sampling collocation points, illustrated on the example of compressible fluids. By changing $p(t, \boldsymbol{x})$ to the distribution over the particle positions in the fluid we effectively change the loss functional in Eq. 6. We then generalize to other settings, such as thermodynamics, by interpreting a positive, scalar quantity of interest with a finite integral as a particle density. Within this work we specifically focus on PDEs that can be derived based on local particle interactions or can be shown to be equivalent to such a view, as for example is the case for the heat equation with its connection to particle diffusion. Notably, we do not require the introduction of Lagrangian updates, as classical mesh-free methods do, which would be based on evaluating the PDE with respect to moving particles (see also section 2).

**Main contributions.**    The main contributions of this paper are as follows:

- We demonstrate that PINNs with uniform sampling strategies (and refinement methods based on uniform proposals) fail in settings with spatially sparse signals as well as in unbounded signal domains; these problems can severely degrade the network's predictive performance.

- In order to overcome these limitations of existing approaches, we propose a truly mesh-free version of Eulerian PINNs, in which the collocation points are sampled using physics-motivated MCMC methods. By staying within the Eulerian framework, we avoid conceptual challenges of classical mesh-free methods based on Lagrangian updates such as the enforcement of boundary conditions.

- The proposed model is applicable to a huge range of dynamical systems governed by PDEs that share an underlying microscopic particle description, such as several hydrodynamic, electro- and thermo-dynamic problems.

- We rigorously evaluate and compare our proposed method with existing approaches in high-dimensional settings. Compared to existing mesh refinement methods, significantly fewer collocation points are required to achieve similar or better predictive performances, while still being more flexible.

## 2 RELATED WORK

**Mesh-Free Fluid Dynamics.** Classical mesh-free approaches in computational fluid dynamics are based on non-parametric function representations, with Smoothed Particle Hydrodynamics (SPH) (Lind et al., 2020; Gingold & Monaghan, 1977) being the most prominent example. In SPH, fluid properties such as the density and pressure are represented by a discrete set of particles and interpolated using a smoothing kernel function. For updating the function forward in time, the particles have to be propagated according to the Lagrangian formulation of the PDE, relying on the kernel for computing spatial derivatives. One of the benefits of such a representation is that mass is conserved by construction. However, Lagrangian updates become challenging when enforcing boundary conditions, requiring the introduction of ad-hoc "dummy" or "mirror" particles (Lind et al., 2020). Instead, we present a mesh-free, particle-based, PINN that does not require Lagrangian updates, and is already applicable in the Eulerian formulation. It should be noted that the proposed pdPINNs can in principle be combined with Lagrangian updates such as proposed by Raissi et al. (2019) and later by Wessels et al. (2020). But as the intention of this work is to improve upon current Eulerian PINNs, we refer to future work for the comparison and extension to the Lagrangian formalism.

**Alternative Meshes and Losses for PINNs.** Recent work proposes local refinement methods for PINNs by adding more samples within regions of high error (Lu et al., 2021; Tadiparthi & Bhattacharya, 2021). Residual adaptive refinement (RAR) is suggested by Lu et al. (2021), which is based on regularly evaluating the PDE loss on a set of uniformly drawn samples. The locations corresponding to the highest PDE loss are then added to the set of collocation points used in training. Tadiparthi & Bhattacharya (2021, preprint) further enhance RAR by learning a linear map between the uniform distribution and the distribution over the PDE loss by optimizing an optimal transport objective. By sampling uniformly and subsequently transforming these samples, it is attempted to focus on regions of higher error. Due to the conceptual similarity to RAR, we will denote this method as "OT-RAR". The work of Nabian et al. (2021) explores Importance Sampling based on the (unnormalized) proposal distribution $||f_\Theta(t, \boldsymbol{x})||_2^2$ for a more sample efficient evaluation of Eq. 6. Samples are drawn using a variation of Inverse Transform sampling (Steele, 1987).

However, in all these cases the underlying mechanism for exploring regions of high error is based on (quasi-) uniform sampling within the boundaries. As such, they do not resolve the issues of unknown boundaries and will furthermore be infeasible in higher dimensions.

**Kinetic Theory: From particles to PDEs.** Kinetic theory shows that essential conservation laws of fluids can be derived from a microscopic (or molecular) viewpoint (Born & Green, 1946). Interactions describing the dynamics of a fluid are described starting from a set of individual particles. The basis of this approach is the so-called *molecular distribution function* $\Psi$ over phase space, i.e. $\Psi(t, \boldsymbol{x}, \boldsymbol{v})$ such that

$$\int_{\Delta \boldsymbol{x}} \int_{\Delta \boldsymbol{v}} \Psi(t, \boldsymbol{x}, \boldsymbol{v}) d\boldsymbol{v} d\boldsymbol{x} \tag{7}$$

is the probability that a molecule with a velocity within $\Delta v = \Delta v_1 \Delta v_2 \Delta v_3$ occupies the volume $\Delta \boldsymbol{x} = \Delta x_1 \Delta x_2 \Delta x_3$. Based on this distribution function, it is possible to define common quantities

as the (mass or particle) density, (local mean) velocity, and macroscopic PDEs by considering the local interactions of individual particles. The one-particle phase space is commonly known from its application in the Boltzmann equation for modelling two-body interactions describing gases (Green, 1956) and active matter (e.g. flocks of birds) (Bertin et al., 2006). The more general form including higher interaction terms is necessary for deriving conservation laws of liquids (Born & Green, 1946).

## 3 PARTICLE-DENSITY PINNS

In this section we introduce the concept of mesh-free *particle-density PINNs (pdPINNs)*. Firstly, we examine limitations of the common PDE loss in Eq. 6 and, secondly, we present a solution by integrating over the position of particles instead of the full support of the signal domain.

The underlying assumption of our approach is that the dynamics described by the PDE can be explained in terms of local interactions of particles. This is the case, for instance, for commonly considered dynamics of gases, liquids or active particles (Hoover & Hoover, 2003; Toner & Tu, 1995).

**Existing limitations of Eulerian PINNs.**   Consider the problem of modeling a (possibly non-steady) compressible fluid, i.e. a fluid with a spatially and temporally evolving density $\rho(t, \boldsymbol{x})$ and velocity $\boldsymbol{v}(t, \boldsymbol{x})$. For the sake of notational brevity, we will denote these by $\rho$ and $\boldsymbol{v}$. Given noisy observations, our particular interest lies in the prediction of particle movements, hence in the approximation of the density (and potentially other physical quantities) with a neural network $\rho_\Theta$. Additional quantities such as the velocity or pressure might also be observed and modeled.

Commonly, the PDE then serves as a physics-based regularizer of the network by enforcing the PDE loss $L_f$ in Eq. 6 during standard PINN training. For this, $L_f$ is evaluated on a set of collocation points that are, for example, uniformly distributed on a bounded region. However, the limitations of this approach already become apparent when considering a simple advection problem defined by the following PDE:

$$\partial_t \rho + \boldsymbol{v} \cdot (\nabla \rho) = 0. \tag{8}$$

Figure 1 illustrates a one-dimensional case on the domain $[0, T] \times \Omega$, with $\Omega = \mathbb{R}$, and a known constant velocity $v \propto 1$. We measure the density $\rho^{(i)}$ at different (spatially fixed) points in time and space $\{(t^{(i)}, \boldsymbol{x}^{(i)})\}$, on which a neural network $\rho_\Theta(t, \boldsymbol{x})$ is trained. For optimizing the standard PDE loss $L_f$ as given in Eq. 6, we would require a bounded region $\Omega_\mathcal{B} := [a, b] \subset \Omega$ with $a < b$ and $a, b \in \mathbb{R}$. This, in turn, leads to two issues:

1. Since the moving density occupies a small subset of $\Omega$, uniformly distributed collocation points within $\Omega_\mathcal{B}$ will enforce Eq. 8 in areas with low-density. This results in insufficient regularization of $\rho_\Theta$.

2. Defining a suitable bounded region $\Omega_\mathcal{B}$ requires a priori knowledge about the solution of the PDE, which is generally not available. Choosing too tight boundaries would lead to large parts of the density moving out of the considered area $\Omega_\mathcal{B}$. Too large boundaries would instead lead to poor regularization as this would worsen the sparsity problem in issue (1.).

In practice, most Eulerian PINNs approaches opt for naively defining a sufficiently wide region $\Omega_\mathcal{B}$, resulting in a poor reconstruction. In the context of our advection problem, this is showcased in Figure 1b. To properly resolve the aforementioned issues, one should (i) focus on areas that have a relevant regularizing effect on the prediction of $\rho_\Theta$ and (ii) adapt to the fluid movements without being restricted to a predefined mesh.

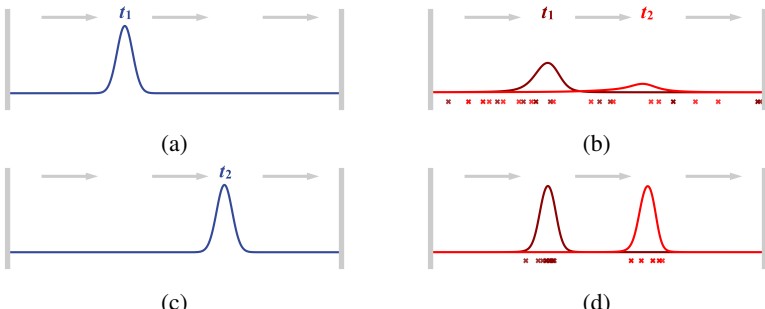

Figure 1: Advection experiment in 1D: (a) ground truth at time $t_1$ and (c) time $t_2$, (b) density prediction with uniform collocation points and (d) particle-density-based collocation points for $t \in \{t_1, t_2\}$, with crosses indicating sampled points.

**Mesh-Free Eulerian PINNs.**    We thus propose to reformulate the PDE loss in Eq. 6 as the expectation of $||f_\Theta(t, \boldsymbol{x})||_2^2$ with respect to the molecular distribution $\Psi(t, \boldsymbol{x})$ introduced in the related work section 2:

$$L_{\text{pd}}(\Theta) \approx \int_{t=0}^{T} \int_{\Omega} \Psi(t, \boldsymbol{x}) \left[ ||f_\Theta(t, \boldsymbol{x})||_2^2 \right] d\boldsymbol{x} \, dt. \tag{9}$$

This completely removes the need of defining ad-hoc boundaries while providing the ability to flexibly focus on highly relevant regions, i.e. those that are more densely populated. As the particle density corresponds directly to the occupation probability of a molecule $\Psi(t, \boldsymbol{x})$ with a changed normalization constant, we can estimate $L_{\text{pd}}$ via samples drawn from the normalized particle density, which is denoted as $\rho_N$. For homogeneous fluids, this coincides with the normalized mass density.

In summary, we propose to draw collocation points from the normalized density:

$$(t_i, \boldsymbol{x}_i) \sim \rho_N(t, \boldsymbol{x}) = \tfrac{1}{Z} \rho(t, \boldsymbol{x}). \tag{10}$$

The true particle positions and the density $\rho_N$ are however unknown in practice. Instead, we have to rely on the learned density $\rho_\Theta(t, \boldsymbol{x})$ as a proxy provided by the neural network. We denote the associated normalized PDF by $q_\Theta(t, \boldsymbol{x}) = \frac{1}{Z'} \rho_\Theta(t, \boldsymbol{x})$ with support on $[0, T] \times \Omega$. The PDE loss is then defined as the expectation w.r.t. $q_\Theta(t, \boldsymbol{x})$:

$$L_{pd}(\Theta) = \mathbb{E}_{q_\Theta(t, \boldsymbol{x})} \left[ ||f_\Theta(t, \boldsymbol{x})||_2^2 \right] = \int_{t=0}^{T} \int_{\Omega} q_\Theta(t, \boldsymbol{x}) \, ||f_\Theta(\boldsymbol{x}, t)||_2^2 \, d\boldsymbol{x} \, dt. \tag{11}$$

In order to approximate this integral, samples need to be drawn from $q_\Theta(t, \boldsymbol{x})$. This can be done in a principled way by using dynamic Monte Carlo methods, despite the fact that the normalization constant $Z$ is unknown. We highlight that, in contrast to the mesh-based loss in Eq. 6, the loss in Eq. 11 is also suitable for problems on unbounded domains such as $\Omega = \mathbb{R}^d$.

**Applicability of pdPINNs.**    Although motivated in the context of an advection problem, the proposed approach is generally applicable to a wide range of PDEs. The advection equation 8 can be seen as a special case of mass conservation (assuming $\nabla \cdot \boldsymbol{v} = 0$), which is one of the fundamental physical principles expressed as a *continuity equation*. This continuity equation relates temporal changes of the fluid density $\rho$ to spatial changes of the flux density $\rho \boldsymbol{v}$ through

$$\partial_t \rho + \nabla \cdot (\rho \boldsymbol{v}) = 0. \tag{12}$$

Another common physical process that is suited for our approach is diffusion, such as in the Heat Equation, where local interactions of particles give rise to the following PDE (as established by Fick's second law):

$$\partial_t T - \alpha \nabla^2 T = 0, \tag{13}$$

where $T$ denotes the temperature interpreted as density, $\alpha$ the thermal (or mass) diffusivity, and $\nabla^2$ the Laplacian operator. By introducing additional constraints to the diffusion and mass-conservation, one can describe viscous fluids with the Navier-Stokes equations or even self-propelled, active particles, for which Toner and Tu (Toner & Tu, 1995; Tu et al., 1998; Toner & Tu, 1998) introduced

hydrodynamic equations. Other possible applications involve Maxwell's equations for conservation of charge in electrodynamics, as well as the distribution of Brownian particles with drift described by the Fokker-Planck equations. In general, our method is applicable in settings where (i) a non-negative scalar field (with a finite integral) of interest can be interpreted as a particle density, and (ii) the local interactions of these particles give rise to the considered PDEs.

## 4 Model and Implementation

A wide range of different network architectures and optimization strategies for PINNs have emerged. They emphasize well-behaved derivatives with respect to the input domain (Sitzmann et al., 2020), allow higher expressivity for modelling high frequency data (Tancik et al., 2020; Wang et al., 2021b), or resolve gradient pathologies within PINNs (Wang et al., 2021a). As our method does not rely on a specific architecture, any such improvement can be easily combined with the proposed pdPINNs. For the experiments in this submission we will use simple fully-connected networks with sinusoidal (Sitzmann et al., 2020) or tanh activations (see section 5).

**Finite total density.**   For reformulating the predicted density $\rho_\Theta$ as a probability, we have to ensure non-negativity as well as a finite integral over the input domain $\Omega$. Non-negativity can for example be achieved via a squared activation function after the last layer. An additional bounded activation function $g$ is then added, which guarantees the output to be within a pre-specified range $[0, c_{max}]$. The integral $\mathbb{R}^d$ can then be enforced to be finite by multiplying the bounded output with a Gaussian kernel. Summarizing these three steps, let $\tilde{\rho}_\Theta$ denote the output of the last layer of our fully connected neural network and $p_{\text{gauss}}(\boldsymbol{x}) = \mathcal{N}(\boldsymbol{x}; \mu, \Sigma)$, then we predict the density $\rho_\Theta$ as

$$\rho_\Theta(t, \boldsymbol{x}) = p_{\text{gauss}}(\boldsymbol{x}) \, g(\tilde{\rho}_\Theta(t, \boldsymbol{x})^2) \leq c_{\max} p_{\text{gauss}}(\boldsymbol{x}). \tag{14}$$

In practice, the choice of $c_{\max}$ does not affect the model as long as it is sufficiently large. The used mean $\mu$ and covariance $\Sigma$ are maximum likelihood estimates based on the observations $\boldsymbol{x}$, i.e. the sample mean $\bar{\boldsymbol{x}}$ and covariance $\bar{\Sigma}$ of the sensor locations. To allow more flexibility in the network, we add a scaled identity matrix to the covariance $\Sigma = \bar{\Sigma} + c \cdot I$, which can be set to a large value for solving PDEs when only initial conditions, but no observations, are available.

**Markov chain Monte Carlo (MCMC) sampling.**   Finally, MCMC methods allow us to draw samples from the unnormalized density $\rho_\Theta(t, \boldsymbol{x})$. We consider several MCMC samplers and emphasize that the wide range of well-established methods offer the ability to use a specialized sampler for the considered problem, if the need may arise. Gradient-based samplers such as Hamiltonian Monte Carlo (Duane et al., 1987; Betancourt, 2017) are particularly suited for our setting, as the gradients of $\rho_\Theta$ with respect to the input space are readily available. For problems where boundaries are known and we have to sample from a constrained region, a bijective transformation is used so that the Markov chain may operate in an unconstrained space (Parno & Marzouk, 2018). In our experience, both Metropolis Hastings and Hamiltonian Monte Carlo already worked sufficiently well for a wide range of PDEs without requiring much fine-tuning. We highlight that pdPINNs do not directly depend on MCMC as a sampler, and alternative sampling methods such as modern variational inference schemes (Rezende & Mohamed, 2015) can also be directly used as a substitute.

For details regarding the samplers used and implementation we refer to the Experiments section 5 and Appendix section A.1.

## 5 Experiments

In this section we demonstrate the advantages of pdPINNs compared to *uniform sampling*, *importance sampling* (Nabian et al., 2021) as well as the adaptive refinement methods *RAR* (Lu et al., 2021) and *OT-RAR* (Tadiparthi & Bhattacharya, 2021). Despite the term *uniform* sampling, we rely in all our experiments on quasi-random Sobol sequences for more stable behavior in the low samples regime. To guarantee a fair comparison, we considered slight variations of the proposed implementations of RAR and OT-RAR, so that only a limited number of collocation points are used. For the pdPINNs we consider multiple MCMC schemes, including inverse transform sampling (IT-pdPINN), Metropolis-Hastings (MH-pdPINN), and Hamiltonian Monte Carlo (HMC-pdPINN) methods.

The models in sections 5.1 and 5.2 are implemented in PyTorch (Paszke et al., 2019), with a custom Python implementation of the MH and Inverse Transform samplers. For the Fokker-Planck experiment in section 5.3, we make use of the efficient MCMC implementations provided by TensorFlow probability (Abadi et al., 2016; Lao et al., 2020) and the utilities of the DeepXDE library (Lu et al., 2021). More details, as well as further experiments comparing the wall-time of the various samplers, are provided in the Appendix with the code being provided in the supplementary material.

## 5.1 Mass conservation for simulated particles

As a challenging prediction task we consider a setting motivated by the real world problem of modelling bird densities and velocities measured from a set of weather radars (Dokter et al., 2011; Nussbaumer et al., 2019; 2021) – or more generally the area of radar aeroecology. A non-steady compressible fluid in three dimensions is simulated by propagating fluid parcels through a pre-defined velocity field, i.e. the fluid is simulated using the conservation of mass as the underlying PDE (see Eq. 12). To provide the network with training observations, we introduce a set of spatially fixed sensors (comparable to *radars*) which count over time the number of fluid parcels within a radius $r$ and over 21 contiguous altitude layers. Another disjoint set of sensors is provided for the validation set while the test performance is evaluated on a grid. The birds-eye view of the setting is shown in Figure 2a, where circles indicate the area covered by the radars. Figure 2b additionally shows the 3D simulated data projected along the $z$-axis and over time. In the Appendix section A.3 we describe the data generation and training setting in detail and provide the corresponding code in the supplementary.

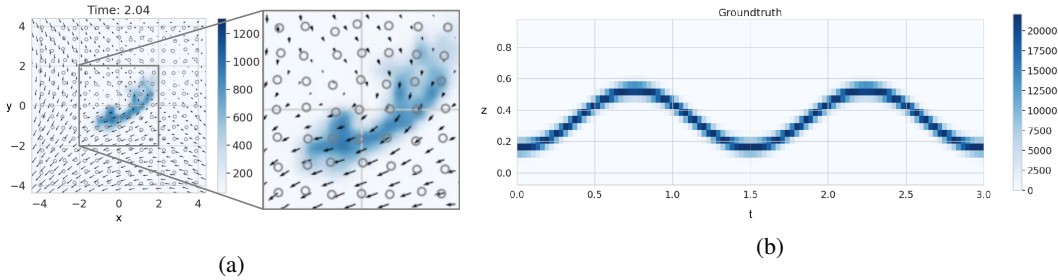

(a)

(b)

Figure 2: Visualization of the 2D compressible fluid experiment. (a) Bird-eye view of the ground truth particle density. (b) $z$-projection of the density over time, obtained summing over the $xy$ grid cells.

For modeling the density and velocity, two sinusoidal representation networks (SIREN) (Sitzmann et al., 2020) $\rho_{\Theta_1}(t, \boldsymbol{x})$ and $\boldsymbol{v}_{\Theta_2}(t, \boldsymbol{x})$ are used, which are then regularized by enforcing the continuity equation for the conservation of mass (see Eq. 12). To showcase the sample efficiency of pdPINNs, experiments are performed over a wide range of collocation points (256 to 65536). In each setting the PDE-weights $w_2$ (see Eq. 5) were selected with a grid search based on the highest 1st quartile $R^2$ in a validation set. The resulting box-plots of the test $R^2$ are provided in Figure 3, where the "Baseline" corresponds to training without any PDE loss. The proposed pdPINN approach clearly outperforms alternative (re-)sampling methods across all numbers of collocation points. Already with very few collocation points (512) pdPINNs achieve results that require orders of magnitude more points (32768) for uniform sampling. Finally, we observe that the performance gap shrinks as the number of collocation points increases, eventually converging to the same limiting value. Even when getting close to the memory limit of a NVIDIA Titan X GPU, other sampling strategies at best achieve comparable results with pdPINNs. In the Appendix (Figure A.6) we provide an additional qualitative comparison of the mass conservation between OT-RAR and MH-pdPINN 2048 samples.

As an additional experiment we simplified the setting by projecting the data onto the $xy$-axis, i.e. the birds-eye view, which is a common setting for geostatistical data (e.g. in Nussbaumer et al. (2019)). The results in this 2D setting, which are provided in the Appendix (Figure A.8) and described in details in section A.3, are very similar in nature to the 3D setting, although with a smaller performance gap with respect to alternative sampling methods. This decrease of the gap is to be expected, as the lower dimensional space is much easier to explore with uniform proposals.

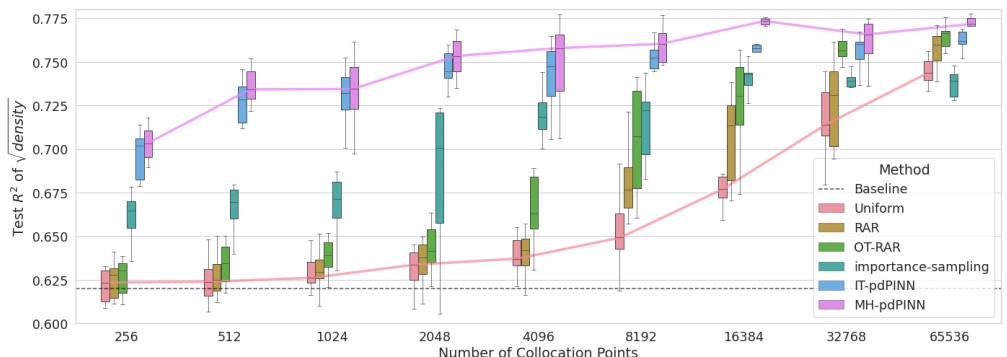

Figure 3: Explained variance of $\sqrt{\rho}$ evaluated on the test set, for different number of collocation points for the 3D mass conservation experiment and for 10 different seeds.

## 5.2 HEAT EQUATION

We further consider a 2D diffusion problem, namely the heat equation introduced in section 3, where randomly distributed sensors provide measurements of the temperature. We focus on a general setting with the initial conditions being zero temperature everywhere except for a specified region, as shown in Figure 4a, and we let the system evolve for $t \in [0, 0.2]$. The networks are only provided sensor measurements of the temperature; for further details see the Appendix section A.4.

Temperature predictions for PINNs with uniform sampling and pdPINNs are illustrated in Figure 4b and 4c, respectively, with the ground truth in Figure 4a. We can observe that the uniform sampling strategy does not allow to focus on the relevant parts of the domain, i.e. regions with high temperature, and that it visibly fails to reconstruct the temperature profile. In contrast, the pdPINN promotes sampling in regions of higher density and predicts the true temperature more reliably. We also evaluate quantitatively the performance of the two approaches in terms of the $R^2$ test error over the predicted temperature and illustrate the results in the Appendix section A.4, where we again observe the same convergence between uniform sampling and pdPINNs for high numbers of collocation points.

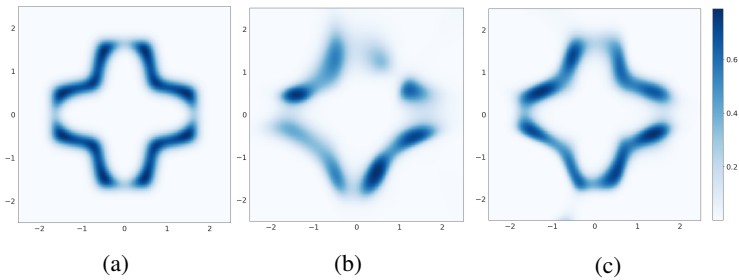

| (a) | (b) | (c) |

Figure 4: Temperature predictions of the heat equation experiment (trained with 128 collocation points) at time $t \sim 0.044$. (a) Ground truth (b) uniform sampling, and (c) pdPINN.

## 5.3 FOKKER-PLANCK EQUATION

For a demonstration of a forward problem, i.e. a setting without any observed data but only initial conditions, we solve the Fokker-Planck (FP) equations in a setting where an analytical solution is available (cf. Särkkä & Solin (2019)). The FP equations describe the evolution of the probability density of the movement of Brownian particles under a drift. More specifically, assume we are given particles at time $t_0$, which are distributed according to $p(t_0, x)$. Let the movements of these particles be described by the following stochastic differential equation, where $W_t$ denotes the standard Wiener process:

$$dX_t = \mu(t, X_t)\, dt + \sigma(t, X_t)\, dW_t \tag{15}$$

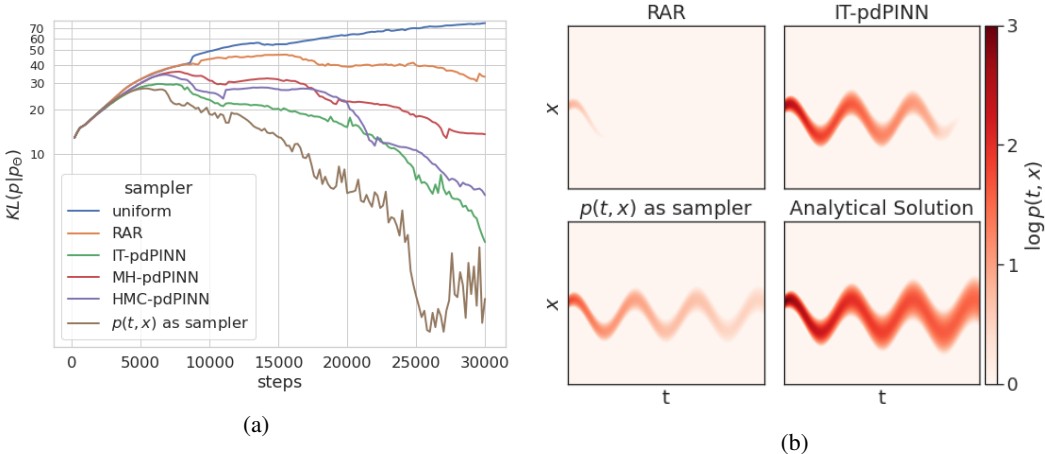

(a)

(b)

Figure 5: Fokker-Planck equation in 1D. (a) KL divergence between the true target distribution and approximation during training, (b) predicted $\log p_\Theta(t, x)$ after training, cropped to $x \in [-0.5, 0.5]$.

with known drift $\mu(X_t, t)$ and diffusion coefficient $D(X_t, t) = \sigma^2(X_t, t)/2$. The FP equation for the probability density $p(t, x)$ of the random variable $X_t$ is then given by

$$\frac{\partial}{\partial t}p(t, x) = -\frac{\partial}{\partial x}\left[\mu(t, x)p(t, x)\right] + \frac{\partial^2}{\partial x^2}\left[D(t, x)p(t, x)\right]. \tag{16}$$

We train a network to predict the (probability) density $p_\Theta(t, x)$ given a known sinusoidal drift and constant diffusion, which are discussed in detail in the Appendix. Data is only provided for the initial condition, and the PDE loss is based on Eq. 16 within the space $\Omega = [-.1.5, 1.5]$ and time $t \in [-1, 1]$. As the analytical solution is available in form of a probability density, we can estimate the KL divergence $KL(p\|p_\Theta)$ to evaluate the performance. Furthermore, we can sample collocation points from the true particle distribution $p(t, x)$ (referred to as "$p(t, x)$ as sampler"), offering a "best case scenario" of pdPINNs. A total of 5000 collocation points were used, and weights were manually tuned based on the error on a validation set. Figure 5a shows the evolution of KL divergence during training, highlighting that pdPINN based methods require fewer steps to achieve a low divergence. In addition, sampling from the true particle distribution leads to the fastest improvement and the lowest divergence after 30000 training steps. A qualitative comparison of the results is given in Figure 5b, showing that RAR and uniform sampling fail to propagate the sine wave forward. The ground truth of the problem and wall-times for different methods are given in the Appendix section A.5.

## 6 CONCLUSION

In this work, we introduced a general extension to PINNs applicable to a great variety of problem settings involving physics-based regularization of neural networks. In order to overcome the limitations of classical mesh-based Eulerian PINNs, we introduce a novel PDE loss that is defined with respect to the particle density in rather general types of PDEs. By employing MCMC methods to sample collocation points from the density approximated by the network, we derive an efficient and easy-to-implement improvement for providing a more appropriate regularization objective in PINNs. In particular, our new pdPINNs are completely mesh-free, thereby overcoming severe efficiency problems of classical PINNs in high-dimensional and sparse settings. Further, the absence of a mesh allows us to elegantly handle settings with uncertain or unknown domain boundaries.

As we have demonstrated, our method is applicable to a wide spectrum of PDEs, ranging from hydrodynamic flow problems to electro- and thermo-dynamic problems, as well as more general applications of the Fokker-Planck equations.

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

# A  APPENDIX

## A.1  BACKGROUND SAMPLING FOR PDPINNS

At initialization, the network prediction $\rho_\Theta$ is random and thus does not carry any useful information, i.e. sampling from this density would be meaningless. Therefore, we start training the pdPINNs with a warm-up phase in which samples are obtained from a pre-specified background distribution:

$$\boldsymbol{x} \sim p_{bg}(t, \boldsymbol{x}) = p(t)p_{bg}(\boldsymbol{x}|t) \tag{17}$$

with $p(t) = \mathcal{U}(0, T)$. To avoid introducing a mesh, we could rely on the previously estimated Gaussian distribution introduced in Section 4, i.e. $p_{bg}(\boldsymbol{x}|t) = p_{\text{gauss}}(\boldsymbol{x})$. As a second alternative, approach we consider random linear combinations of the convex hull of $\{x^{(i)}\}_{i=1}^N$ spanned by $c$ data points summarized as rows of matrix $Z \in \mathbb{R}^{c \times d}$. This leads to $\boldsymbol{x} = \boldsymbol{m}Z$ with weight $\boldsymbol{m} \in \mathbb{R}^c$ which can be drawn from a Dirichlet distribution, i.e. $\boldsymbol{m} \sim Dir(\boldsymbol{\alpha} = 1)$. Of course, a uniform sampling mechanism on a defined region is also suitable and the definitive choice depends on the data and PDE at hand. However, we found that all of these methods work well in practice.

We initially draw all samples from the background distribution, and then slowly increase the proportion of samples obtained from the particle density, as we found that leaving some background samples slightly helps in the training.

## A.2  IMPLEMENTATION OF RAR AND OT-RAR

For our comparison, we considered the adaptive refinement methods *RAR* and *OT-RAR*, proposed by Lu et al. (2021) and Tadiparthi & Bhattacharya (2021, preprint). Both methods rely on consecutive refinements of a fixed grid in the initial proposal. The number of collocation points is steadily increased and collocation points once added will not be removed. To allow for a fairer comparison, we adapt both methods to use a limited budget of points, and in addition we regularly resample them. This leads to a slightly modified version of the methods which is similar in spirit. For learning the linear mapping proposed by Tadiparthi & Bhattacharya (2021), we rely on the PyOT (Flamary et al., 2021) implementation of Knott & Smith (1984). The pseudo-code for sampling a set of collocation points is given in Algorithm 1 and Algorithm 2. The required input $f_\Theta$ refers to the PDE approximated by the network, as discussed in Section 1. For more specific details on the methods we refer to the original papers.

---

**Algorithm 1** Adapted RAR

**Input:** $f_\Theta$, uniform distribution $\mathcal{U}_\mathcal{B}$,
  number of col. points $k$, previous col. points $X_{\text{prev}}$.

$X_{\text{prop}} \leftarrow [\boldsymbol{x}_1, \boldsymbol{x}_2, \ldots, \boldsymbol{x}_k]^T$ with $\boldsymbol{x}_i \sim \mathcal{U}_\mathcal{B}$   ▷ Sample proposals
$X_{\text{comb}} \leftarrow \text{concat}(X_{\text{prev}}, X_{\text{prop}})$   ▷ Concatenate old and new points
$X_{\text{new}} \leftarrow \text{topk}(X_{\text{comb}}, \ ||f_\Theta(X_{\text{comb}})||_2^2, \ k)$   ▷ Keep top $k$ proposed points based on $f_\Theta$

**Output:** $X_{\text{new}}$

---

## A.3  EXPERIMENTS: CONSERVATION OF MASS

In the supplementary material we provide code in Python for the data generation and for the pdPINN model. Below we provide the details for all the experiments we conducted. Furthermore, we provide short videos showing the predicted density movements for each different approach. More details on this can be found in the README.html provided in the supplementary files.

All experiments were run on a computing cluster using *Nvidia GeForce GTX Titan X GPUs* with 12 GB VRAM. Settings that required more memory were run on a RTX8000 with 48GB VRAM. Up to 16 Titan X GPUs could be used in parallel, or 4 RTX8000. In most settings, training in each experiment took less than 10 minutes.

---

**Algorithm 2** Adapted OT-RAR

---

**Input:** $f_\Theta$, uniform distribution $\mathcal{U}_\mathcal{B}$,
number of col. points $k$,
number of points for empirical distribution $j < 2k$,
previous col. points $X_{\text{prev}}$.

$X_{\text{prop}} \quad \leftarrow [\boldsymbol{x}_1, \boldsymbol{x}_2, \ldots, \boldsymbol{x}_k]^T$ with $\boldsymbol{x}_i \sim \mathcal{U}_\mathcal{B}$ $\qquad\qquad\qquad$ ▷ Sample proposals
$X_{\text{comb}} \quad \leftarrow \text{concat}(X_{\text{prev}}, X_{\text{prop}})$ $\qquad\qquad\qquad$ ▷ Concatenate old and new points

$X_{\text{target}} \quad \leftarrow \text{topk}(X_{\text{comb}}, \, ||f_\Theta(X_{\text{comb}})||_2^2, \, j)$ $\qquad$ ▷ $j$ samples for target empirical distribution
$X_{\text{source}} \quad \leftarrow [\boldsymbol{x}_1, \boldsymbol{x}_2, \ldots, \boldsymbol{x}_j]^T$ with $\boldsymbol{x}_i \sim \mathcal{U}_\mathcal{B}$ $\qquad$ ▷ $j$ samples for source empirical distribution

$M_{\text{OT}} \quad \leftarrow \text{LinOT}(X_{\text{source}}, X_{\text{target}})$ $\qquad$ ▷ Obtain linear operator that maps to target distribution

$X_{\text{new}} \quad \leftarrow [\boldsymbol{x}_1, \boldsymbol{x}_2, \ldots, \boldsymbol{x}_k]^T$ with $\boldsymbol{x}_i \sim \mathcal{U}_\mathcal{B}$ $\qquad\qquad\qquad$ ▷ Sample uniformly
$X_{\text{map}} \quad \leftarrow M_{\text{OT}}(X_{\text{new}})$ $\qquad\qquad\qquad$ ▷ Map samples to target distribution

**Output:** $X_{\text{map}}$

---

### A.3.1 ADDITIONAL EXPERIMENTAL RESULTS

**3D Setting.** Figure A.6 showcases the projection of the density in the onto the z axis for a random run of the OT-RAR method and the Metropolis-Hastings based pdPINN when using 2048 collocation points. The OT-RAR PINN shows disconnected density predictions that clearly violate mass conservation, whereas the Metropolis Hastings based pdPINN is capable of mostly preserving it. The boxplot in Figure A.8 highlights the difference in required number of collocation points of

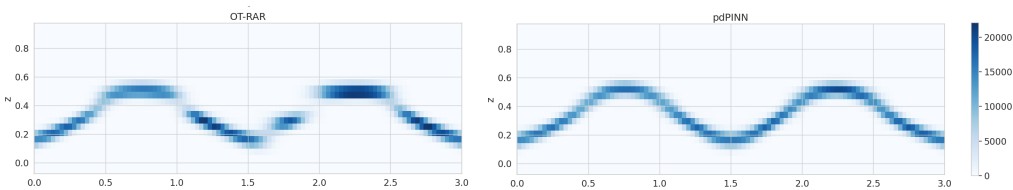

Figure A.6: Mass conservation experiment (3D): Predictions (obtained with 2048 collocation points) summed over $xy$ grid cells to obtain $z$-axis projection over time.

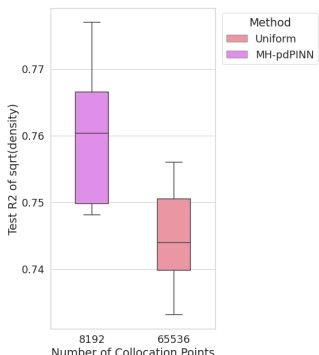

Figure A.7: Mass conservation experiment (3D): Boxplot of test $R^2$ of $\sqrt{\rho}$ comparing pdPINN and uniform sampling with a factor 8 difference for the number of collocation points. For each method, the used PDE weight was selected based on the highest 1st quartile $R^2$ in a validation set.

**2D Setting.** As mentioned in Section 5, we repeated the Conservation of Mass experiment in a slightly altered setting, where the data is projected onto the *xy*-plane, reducing it to a 2D+Time problem. The general setup is similar to the 3D setting, although a smaller network and different training parameters are used, which are listed in the following sections below.

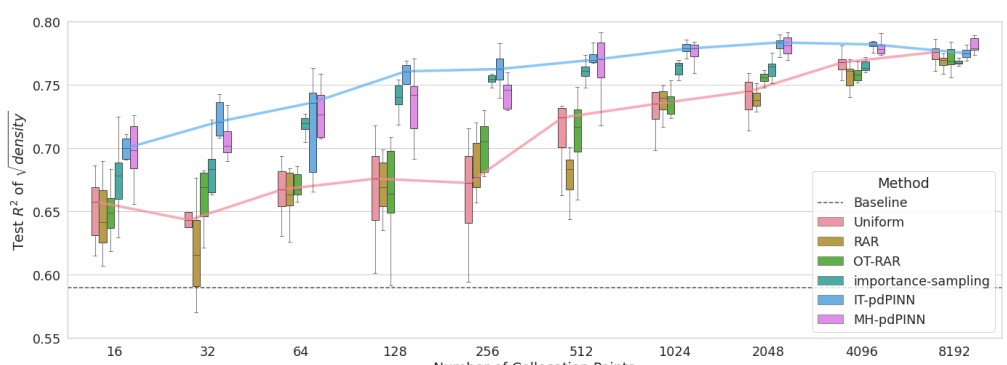

Figure A.8: Explained variance of $\sqrt{\rho}$ evaluated on the test set, for different number of collocation points for the 2D mass conservation experiment.

### A.3.2 DATA GENERATION

Here we provide a more detailed description for the generated data, namely the used velocity field, and the method for obtaining simulated "radar measurements".

**Velocity field.** The velocity field in the $xy$-plane was generated from a scalar potential field $\Phi : \mathbb{R}^2 \to \mathbb{R}$ and the z-component of a vector potential $a : \mathbb{R}^2 \to \mathbb{R}$. Through the Helmholtz decomposition[1] we can construct the velocity field $\boldsymbol{v}_{\mathrm{xy}} : \mathbb{R}^2 \to \mathbb{R}^2$:

$$\boldsymbol{v}_{\mathrm{xy}}\left( \begin{bmatrix} x \\ y \end{bmatrix} \right) = -\nabla\Phi + \begin{bmatrix} \delta a/\delta y \\ -\delta a/\delta x \end{bmatrix}. \tag{18}$$

For both experiments the following fields were used:

$$\Phi\left( \begin{bmatrix} x \\ y \end{bmatrix} \right) = -\frac{1}{2}(x-2)\cdot(y-2), \tag{19}$$

$$a\left( \begin{bmatrix} x \\ y \end{bmatrix} \right) = -\frac{1}{5}\exp\left( -\left(\frac{2}{3}x\right)^2 - \left(\frac{2}{3}y\right)^2 \right). \tag{20}$$

The derivatives were obtained using the symbolic differentiation library SymPy (Meurer et al., 2017). To add a nonsteady component, the resulting velocity field is modulated in amplitude as a function of time $t \in [0,3]$:

$$\boldsymbol{v}_{\mathrm{xyt}}\left( t, \begin{bmatrix} x \\ y \end{bmatrix} \right) = \boldsymbol{v}_{\mathrm{xy}}\left( \begin{bmatrix} x \\ y \end{bmatrix} \right)\left( \frac{3}{2}\left|\sin\left(\frac{2}{3}\pi t\right)\right| + 0.05 \right). \tag{21}$$

The $z$ (altitude) component of the velocity only depends on time and is given by:

$$v_z(t) = 1.6 \cdot \sin\left( \frac{4}{3}\pi t \right). \tag{22}$$

**Simulation.** For the initial distribution of the fluid, the particle positions were drawn from Gaussian mixtures. For $t \in [0,3]$, these particles were simulated using the above constructed velocity field. Overall, the paths of the roughly 240000 parcels were simulated using a basic backward Euler scheme.

---

[1]This is the 2D formulation of the Helmholtz decomposition, where the vector potential has non-zero components only along the $z$-axis as in $\boldsymbol{a}_{\mathrm{3d}} = [0,0,a]^T$. The full decomposition is commonly written as $\boldsymbol{v}_{\mathrm{3d}} = -\nabla\Phi_{\mathrm{3d}} + \nabla \times \boldsymbol{a}_{\mathrm{3d}}$.

**Measurements.** The measurements at the sensors were obtained by counting the number of particles within a given radius over multiple timesteps. The density corresponds to the mass divided by the sensor area, and the velocity is an average over all the particle velocities. For the training data additional zero-mean isotropic Gaussian noise is added to all measurements. In the 3D setting, data measurements of density and velocity are obtained by $13^2$ sensors on the $xy$-plane, within region $[-3, 3]^2$ at 11 equidistant timesteps. In the 2D setting, the same set of sensors is used.

### A.3.3 Architecture and Training

In both experiments, the networks for density $\rho_{\Theta_1}$ and velocity $v_{\Theta_2}$ prediction (parameterized by $\Theta_1$ and $\Theta_2$, respectively) are fully-connected layers with sinusoidal activation functions, as proposed by Sitzmann et al. (2020). The number of layers and units for each setting is shown in Table A.1. The sine frequency hyperparameter required in the SIREN architecture was tuned by hand according to the validation loss of the baseline model (i.e. without a PDE loss), leading to a sine-frequency of 12 for the 2D setting, and 5 for the 3D setting. We note that the proposed default value of 30 in Sitzmann et al. (2020) heavily overfits our relatively low-frequency data and we thus recommend an adjustment of this hyperparameter for usage in PINNs.

For training the network, the ADAM optimizer (Kingma & Ba, 2014) with a learning rate of $8 \times 10^{-4}$ (2D Setting) or $10^{-4}$ (3D Setting) was used. The learning rate was multiplied by a factor of 0.99 each epoch. All models were trained for 300 (3D setting) or 500 (2D setting) epochs. The 2D setting was trained using full-batch gradient descent, whereas for the 3D setting we used a mini-batch size of 6931. In all experiments we trained and evaluated on 10 different random seeds.

Table A.1: Architecture for Particle Simulation Experiments.

| Experiment | Input | Output Variable | # Hidden Layers | # Hidden Units |
|---|---|---|---|---|
| 2D | $[0, T] \times \mathbb{R}^2$ | Density $\rho_{\Theta_1} \in \mathbb{R}_+$ | 2 | 256 |
|  |  | Velocity $v_{\Theta_2} \in \mathbb{R}^2$ | 1 | 64 |
| 3D | $[0, T] \times \mathbb{R}^3$ | Density $\rho_{\Theta_1} \in \mathbb{R}_+$ | 6 | 256 |
|  |  | Velocity $v_{\Theta_2} \in \mathbb{R}^3$ | 3 | 256 |

### A.4 Experiments: Heat Equation

The dataset for the heat equation experiment was generated by numerically solving the heat equation through the finite difference method, precisely the Forward Time, Centered Space (FTCS) approximation (Recktenwald, 2004). We used Dirichlet boundary conditions in form of zero temperature around a squared shape far away from the relevant domain. These boundary conditions are not provided to the PINNs for a slightly more difficult setting. Overall, the dataset is composed of 1000 training points, 1971120 test points and 492780 validation points. We made sure training points contained enough information about the initial condition, i.e. we selected a sufficient amount of points around the initial source of non-zero temperature. In contrast, validation and test points are taken uniformly in time and space. During the warm-up phase of the pdPINN training, collocation points were sampled uniformly, and afterwards 90% of the samples were drawn from the particle density distribution, which is proportional to the modeled temperature. Collocation points were re-sampled every 500 epochs. Differently from previous experiments, the employed architecture is a fully-connected two-layer neural network with 32 hidden units and tanh activations. The implementation is in PyTorch (Paszke et al., 2019), using the ADAM optimizer (Kingma & Ba, 2014) combined with an exponential learning rate scheduler which multiplies the learning rate by a factor of 0.9999 at each epoch, starting with a rate of $10^{-4}$ and decreasing it until reaching a minimum value of $10^{-5}$. Training was terminated through early-stopping, as soon as the validation $R^2$ didn't improve for more than 3000 epochs.

**Additional results.** Figure A.9 illustrates the test $R^2$ of the predicted $T$ averaged over 20 different seeds. Error bars correspond to 95% confidence interval for the mean estimation, based on 1000 bootstrap samples, while colors indicate the different PDE weights $w_2$ explored. As in previous settings, we show that with few samples (16) the regularization enforced by the PDE loss is not strong

enough, leading to comparable results in both approaches (as expected). Hence PINNs and pdPINNs show similar results in this regime. However, as the number of samples increases (32-64-128-256), the PDE loss enforced by the proposed pdPINNs quickly and steadily outperforms uniform sampling. Lastly, we also verified that in the limit of high samples (512-1024) the two sampling strategies converge, as in such a low-dimensional domain the uniform samples fully and densely covers the considered area. This, again, is in line with the observed results of the other experiments.

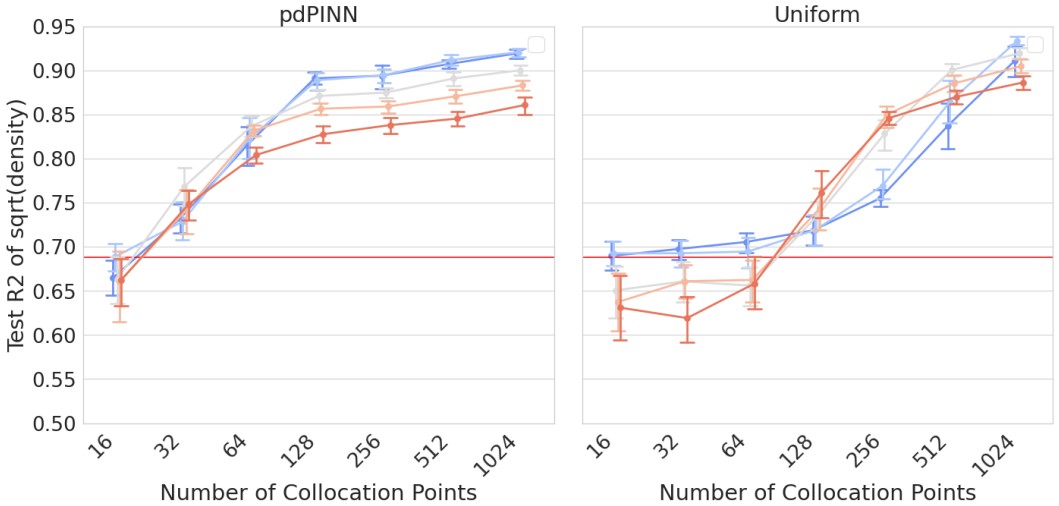

Figure A.9: Test $R^2$ of predicted $T$ in the heat equation experiment as a function of different number of collocation points. Results are averaged over 20 different seeds and the resulting error bars correspond to $95\%$ confidence interval for the mean estimation, based on 1000 bootstrap samples. Different colors indicate different PDE weights $w_2$.

### A.5 Experiments: Fokker-Planck Equations in TensorFlow

Within the Fokker-Planck experiment we showcase the different training behaviors of uniform sampling, RAR, and multiple MCMC samplers. Due to the low dimensionality of the problem, we additionally consider a Inverse-Transform (IT) sampler (Steele, 1987) for efficiently sampling from the density. The IT sampler relies on the empirical cdf estimated via uniform samples drawn over the whole domain. This method does not require building up a Markov Chain, and is thus very fast, but only works well in low dimensions.

More specifically, we compare the following methods for selecting collocation points, with a highly efficient implementation of the MCMC methods provided by TensorFlow probability:

   I.) Uniform sampling

  II.) Residual Adaptive Refinement (Lu et al., 2021)

 III.) pdPINN with Inverse-Transform (IT) sampling (Steele, 1987)

 IV.) pdPINN with Metropolis-Hastings (MH) MC with parallel tempering (Earl & Deem, 2005)

  V.) pdPINN with Hamiltonian MC (HMC) with parallel tempering (Earl & Deem, 2005) and dual averaging step-size adaptation (Hoffman et al., 2014, section 3.2)

#### A.5.1 Setting and Analytical Solution

We consider the following setting over the time interval $[t_0, t_n] = [-1, 1]$ with drift function $\mu$, noise $\sigma$ and initial particle positions $p(x|t = t_0)$ given by

$$\mu(X_t, t) = \mu(t) = \sin(10t) \tag{23}$$
$$\sigma(X_t, t) = \sigma = 0.06 \tag{24}$$
$$p(x|t = t_0) = \mathcal{N}(0, 0.02^2 \cdot I_d) \tag{25}$$

The PDE has an analytical solution (cf. Särkkä & Solin (2019)) which is given by

$$p(x|t) = \mathcal{N}(\mu_s(t), \sigma_s^2(t)) \tag{26}$$
$$p(t) = \mathcal{U}(t_0, t_n) \tag{27}$$
$$\mu_s(t) = -\frac{\cos(10t)}{10} + \frac{\cos(10)}{10} \tag{28}$$
$$\sigma_s^2(t) = 0.0036t + 0.004. \tag{29}$$

For evaluating the deviation of our prediction to the solution, we evaluate the KL divergence between the analytical solution and the network approximation $KL(p(x,t)|\hat{p}_\Theta(x,t))$ by sampling 10000 points from the true $p(x,t)$.

### A.5.2 SETUP

We use a SIREN network and additionally sample (5000) collocation points at the initial time-step, which is the default behavior of DeepXDE. An overview of the architecture and training details is given in Table A.2. Experiments were performed with a NVIDIA GeForce RTX 2080 Ti and an Intel(R) Xeon(R) CPU E5-1660 v3 @ 3.00GHz processor.

Table A.2: Architecture for Fokker-Planck experiments.

| Experiment | Input | Output Variable | # Layers | # Units | col. points | epochs |
|---|---|---|---|---|---|---|
| 1D | $[0,T] \times \mathbb{R}$ | $p_\Theta(x,t) \in \mathbb{R}_+$ | 5 | 64 | 5.000 | 30.000 |

### A.5.3 WALL TIME

The wall times for the different methods are provided in Figure A.10. Although Metropolis-Hastings and Hamiltonian Monte Carlo require more time per step compared to uniform sampling, the used inverse transform sampling achieves a similar speed.

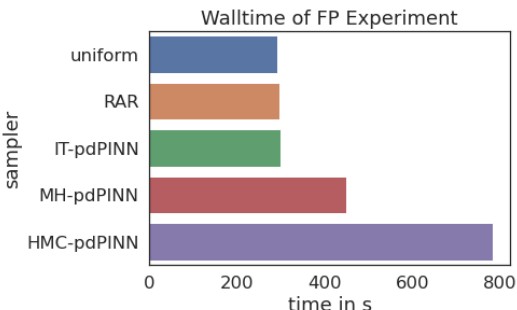

Figure A.10: Total run-times for the Fokker-Planck experiment. Seeds were selected randomly.

