# OpenReview forum: "Mesh-free Eulerian Physics-Informed Neural Networks"
_ICLR.cc/2023/Conference — Submitted to ICLR 2023_

### Official Review · Reviewer_ukjY · 2022-10-25

**Confidence:** 3
**Correctness:** 3
**Technical Novelty And Significance:** 4
**Empirical Novelty And Significance:** 2
**Recommendation:** 6

**Clarity, Quality, Novelty And Reproducibility:**

The manuscript is nicely written. The code is attached. I can reproduce some of the results.

**Strength And Weaknesses:**

Strength:
- This manuscript is well-written.
- The proposed method is easy to follow: it is true that the sampling matters in training PINNs.
- The attached code showed good reproducibility.
- The experiments validating this method are extensive.

Weaknesses:
- There are multiple algorithmic choices not justified in the manuscripts. For example, why use the square as the non-negative translator instead of the absolute? More ablation studies are expected to examine them.

**Summary Of The Paper:**

This manuscript proposes an interesting idea based on the observation that PINNs are normally trained using uniform sampling, which might not be the optimal case. Using the proposed method, the region of interest will be naturally sampled with a higher density. This manuscript applied their method to a set of common PDE problems, including Fokker-Planck equations.

**Summary Of The Review:**

This manuscript builds its idea based on an interesting observation. However, there are ablation studies that need to justify their choices and provide more advice to practitioners.

---

> ### Author Response · Authors · 2022-11-18
> **Response to the Review**
>
> We thank the reviewer for the feedback, and for taking the time to reproduce some of the results.
>
> >There are multiple algorithmic choices not justified in the manuscripts. For example, why use the square as the non-negative translator instead of the absolute?
>
> The choice of the non-negative transformation does not affect our proposed sampling method in any meaningful way, it is just required to output a non-negative quantity.
> Taking the absolute value, as suggested by the reviewer, or an exponential transformation would also be possible.
> The definite choice has to be done depending on the density that is modeled
> and is more of a Neural Network architecture question, unrelated to our proposed pdPINNs.
>
> >More ablation studies are expected to examine them.
>
> In our experiments, we consider a wide range of different hyperparameters and samplers that we consider important for the proposed method and the performance of the PINNs.
> We would like to highlight that this already amounted to multiple thousand different runs.
>
> We provide experiments comparing the performance when substituting or completely removing the density-based samplers, and compared it to various alternative resampling methods available in the PINN literature.
>
> Concerning further ablation studies, we are not sure which other ablations would provide additional information about our model.

---

### Official Review · Reviewer_J4Gd · 2022-10-31

**Confidence:** 4
**Correctness:** 2
**Technical Novelty And Significance:** 2
**Empirical Novelty And Significance:** 1
**Recommendation:** 3

**Clarity, Quality, Novelty And Reproducibility:**

The motivation of the problem and approach are well explained. However, the use of a different loss function for PINN that is sampled from MCMC is not a novel solution to the main issue identified. The experiments also do not support the resolution of the problem mentioned. The choice of the statistics for the loss function is arbitrary and should have been better justified. The hyperparameter choices especially the comments on choosing c in equation 14 are not well justified and reproducible.

The paragraphs of the Main contributions 2 is not sufficiently justified. How do the authors avoid the conceptual challenges of classical mesh-free methods? A discussion is missing and experiments do not show this point. The third point is not a contribution and not supported by experiments as well.



**Strength And Weaknesses:**

The strength of the paper is the realization that existing PINNs with uniform sampling of the solution space is not sufficient for several applications and clearly discussed with the advection equation. The use of the expected molecular distribution function is also interesting.

The weaknesses of the paper are as follows:
1. Why is the expected value of molecular distribution used? In the third application, the KL divergence is used for performance evaluation. Why not use that for the training itself. Are there any mathematical arguments/proofs on why training for expected distribution is good. For the hyperparamenter tuning, the quartile values are used. The choice seems arbitrary and not well explained or justified.

2. The advection equation issue is well explained. So the reviewer (and any reader) will expect that the advection problem will be shown in the applications, at least in the appendix. The chosen application in section 5.1 seems to be a different version (not a soliton that struggles with PINNs).

3. MCMC (or any variant including HMC) itself is too expensive to sample from. The authors also allude to the memory requirements in their high-end GPUs. Is MCMC the right approach for this problem? Is it not making the problem worse? Now if not expectation and some other information metric is used, the cost will go up more!


**Summary Of The Paper:**

The paper proposes the use of the expected molecular distribution function as the loss function to train a Physics Informed Neural Network. Then, the authors use MCMC to sample the molecular distribution function to compute the expectation using the output of the neural network for the molecular distribution function. The authors show three applications: mass conservation for simulated particles, heat equation and fokker planck equation (stochastic differential equation).

**Summary Of The Review:**

Overall, the paper lacks a good justification for the loss function choice and improvements to the learning representation this choice brings to the table. The experiments do not justify the use of the new loss function and the advantages are not sufficiently highlighted. Overall the idea of the paper is not a significant improvement in the field.

---

> ### Author Response · Authors · 2022-11-18
> **Response to the Review (1/2)**
>
> We thank the reviewer for the comments, but notice that there might be some misunderstandings of our work which we can hopefully clarify. Specifically, we do include experiments for the introductory advection problem, although in higher dimensional settings.
>
> The 1d advection equation is the continuity equation with a constant velocity $v(x, t) = c$ with $c \in \mathbb{R}$:
>
>
> \begin{align}
> \frac{\partial \rho}{\partial t} + \nabla \cdot (\rho {v}) &= 0 \\
> \nabla \cdot (\rho {v}) &= \frac{\partial (\rho {v})}{\partial x} = {v} \frac{\partial \rho}{\partial x} +\rho \underbrace{\frac{\partial v}{\partial x} }_{=0}\\
> \Rightarrow \frac{\partial \rho}{\partial t} + v \cdot (\nabla \rho) &= 0
> \end{align}
>
> More generally for $\boldsymbol{v}: \mathbb{R}^d \rightarrow \mathbb{R}^d$, the advection equation comes from the continuity equation when considering divergence-free velocities, i.e. if we assume $\nabla \cdot \boldsymbol{v} = \boldsymbol{0}$:
> $$ \frac{\partial \rho}{\partial t} + \nabla \cdot (\rho \boldsymbol{v}) =  \frac{\partial \rho}{\partial t} + \boldsymbol{v} \cdot (\nabla \rho) + \underbrace{\rho (\nabla \cdot \boldsymbol{v}) }_{=0} = \frac{\partial \rho}{\partial t} + \boldsymbol{v} \cdot (\nabla \rho) = 0
> $$
>
> The second equality follows from the product rule applied to the divergence operator (see e.g. [1]).
>
>
> >  In the third application, the KL divergence is used for performance evaluation. Why not use that for the training itself.
>
> The KL Divergence we use is:
> $$KL(p||p_\Theta) =E_p[log\frac{p(x, t)}{p_\Theta(x, t)}]$$
> We can only evaluate this KL-divergence if we have access to the true probability $p(x,t)$, which is unknown. During training, we only have access to the (unnormalized) $z_{p_\Theta} p_\Theta(x, t)$, and can sample from $p_\Theta$ via for example MCMC.
>
> >The advection equation issue is well explained. So the reviewer (and any reader) will expect that the advection problem will be shown in the applications, at least in the appendix. The chosen application in section 5.1 seems to be a different version (not a soliton that struggles with PINNs).
>
>
> This seems to be a misunderstanding, as the advection equation is actually a special case of the continuity equation that we consider in our more involved experiments.
> Although we do mention that in Section 3 when discussing the applicability of pdPINNs, we will try to write it more clearly to avoid confusion.
>
>
>
> >MCMC (or any variant including HMC) itself is too expensive to sample from.
>
>
> Efficient implementations of basic MCMC samplers are actually very fast and do not necessarily lead to a large slow-down.
> We refer to Figure A.10 in the Appendix, where the total runtimes of different samplers are shown.
> Please note that even if run-time is an issue, one can just rely on inverse transform sampling, which is just as fast as uniform sampling, but still outperforms it in all experiments.
>
>
> >The authors also allude to the memory requirements in their high-end GPUs.
>
>
> The proposed method actually helps us to avoid large memory requirements, as we have a much higher sampling efficiency.
> We only need to allocate to the GPU memory the points that we will actually be using as collocation points during training.
> We do not need to keep track of the whole trace of the MCMC proposals.
>
> Specifically, we want to highlight Figure A.7 in the Appendix, which shows that our pdPINNs with ~8k samples already outperform uniform sampling with ~65k samples.
> With pdPINNs we only passed 8k collocation points to the network for each training iteration (and consequently only needed to allocate for these 8k).
> With Uniform sampling we need to pass 65k collocation points to the network for each training iteration, which is how we got to the memory limit of our GPU.
>
> Nonetheless, our methods do not require high-end GPUs, and many experiments can (and actually were) run on the CPU. But as we have a very extensive evaluation of many thousand runs over different configurations, we used a cluster.
>
> ### References
>
> [1] https://en.wikipedia.org/wiki/Divergence#In_arbitrary_finite_dimensions

---

> > ### Author Response · Authors · 2022-11-18
> > **Response to the Review (2/2)**
> >
> >
> >
> > > Is MCMC the right approach for this problem? Is it not making the problem worse?
> >
> >
> > With our work, we propose to sample from the density, but do not pose any restrictions on how to do that.
> > MCMC is one way to do that, and we decided for it due to it's theoretical guarantees.
> > In practice, other approaches can be more efficient, which is why we also included the very simple Inverse Transform sampling.
> >
> >
> >
> > In general, any method for sampling from an unnormalized density might be used.
> > Other examples might include the whole area of variational inference, be that with normalizing flows or simple mixtures of Gaussians.
> > Another reviewer also suggested the use of sequential Monte Carlo.
> >
> > But as the area of sampling methods is an extremely extensive field we do not wish to evaluate all of them, but rather show that the general idea of sampling from the density is beneficial when training PINNs.
> > For a specific problem setting, the most appropriate sampler can be chosen by the user.
> >
> > > Why is the expected value of molecular distribution used?
> > > Now if not expectation and some other information metric is used, the cost will go up more!
> >
> > In PINNs the PDE loss is evaluated as the average PDE loss (or sum) overall collocation points, similar to a MSE loss when training a network on data.
> > As an average is the sample-based estimator of an expectation, we are only interested in the expected value.
> >
> > This is a slightly different setting from classical applications of MCMC, where one is actually interested in the distribution or other summary statistics. Instead, we are only interested in estimating expectations.
> >
> > >However, the use of a different loss function for PINN that is sampled from MCMC is not a novel solution to the main issue identified.
> >
> > We are not aware of any previous work proposing to sample from the particle distribution for alleviating the discussed limitations of PINNs. Consequently, we would ask the Reviewer to clarify why our work would not be a novel solution.
> >
> > >The hyperparameter choices especially the comments on choosing c in equation 14 are not well justified and reproducible.
> >
> > The introduced $c$ is more of a formal guarantee that the predicted density is bounded.
> > In practice, it is taken to be a very large value, e.g. 1e12.
> > As this would then not affect the network (assuming we model densities below that value), this is very much reproducible.
> >
> >
> > > How do the authors avoid the conceptual challenges of classical mesh-free methods?
> > The conceptual challenges of these methods are directly avoided by staying the Eulerian framework, where we can enforce boundary conditions similar to any other PINN by directly enforcing them in our network.
> > The issues do not arise due to the mesh-free nature, but rather due to the Lagrangian framework itself.
> >
> > Classical mesh-free methods, such as Smooth Particle Hydrodynamics (SPH) propagate individual particles or parcels forward in time, with the parcels themselves serving as a non-parametric function representation.
> > Enforcing boundary constraints becomes difficult, as changing the function is only possible by changing or introducing particles.
> >
> > In our framework, we still have a parameterized function given by the network and do not need to propagate the particles forward. Instead, we can just sample particles at the next time step by looking at the predicted density.
> >
> > A discussion of this is provided in the related work Section 2.

---

> > ### Comment · Reviewer_J4Gd · 2022-12-05
> > **Response to author's response**
> >
> > In fact, there is no misunderstanding. The reviewer is well aware of the derivation of the advection equation, and the authors have also mentioned it is a special case. However, the concern is that the case that doesn't work efficiently with classical PINN is a soliton (a solitary wave) solution and not really a generic advection solution. The comment was about the test case not adequately doing justice to the soliton claim. Sorry that the authors had to misunderstand a non-existent misunderstanding.
> >
> > Still, the choice of expectation and the arbitrary choice of quantiles is not explained. The authors must try large-scale applications and then comment on whether implementations of MCMC are indeed actually very fast. After reading the other comments, this reviewer is not updating scores.

---

### Official Review · Reviewer_ZR9E · 2022-10-31

**Confidence:** 4
**Correctness:** 4
**Technical Novelty And Significance:** 2
**Empirical Novelty And Significance:** 2
**Recommendation:** 6

**Clarity, Quality, Novelty And Reproducibility:**

$\cdot$ Regarding the clearness, this paper is very clearly written with a lot of details in each aspect.

$\cdot$ The novelty, as stated above, is mostly on the proposal of defining a specific loss function framework than a novel proposal of neural network architecture, later of which could limit the novelty for a paper targeting at the top venue.

$\cdot$ The reproducibility is feasible, and the authors also attached code in the supplementary materials.


**Details Of Ethics Concerns:**

There is no ethics concern in this paper.

**Strength And Weaknesses:**

The strengths of this paper are:

$\mathbf{1}.$ The idea of using particle density and emphasizing locality of interactions in loss function is novel. It provides a rigorous view of defining a specific loss function for general PDE constrained loss function (of neural networks). The proposed method also incorporate a MCMC sampling strategy, which supports the idea very well.

$\mathbf{2}.$ The paper is very well organized and clearly written. The literature review is comprehensive along with arguments and to introduce the motivation of the proposed method. Overall, the paper is easy to read for audience in a broad area of deep learning.

The weakness of this paper:

$\mathbf{1}.$ There lacks novel proposal of network architecture, which might limit the novelty of this paper. Here, it is worthwhile noting that the novel definition of loss function does qualify as a good contribution. However, the architecture of neural network is not novel.

$\mathbf{2}.$ Author discussed the applicability of the propose methods. However, the limitation of applicability seems to be obvious as the authors stated. But, it is rather an unfamiliar area for me.

**Summary Of The Paper:**

This paper proposes a network based method, in the realm of physics informed methods, to solve a class of PDEs. The proposed methods mainly extend the general framework of having PDE as constraints in the loss function by introducing a partial density and locality of particle interactions. It also proposes to use MCMC method to sample the collocation points.

**Summary Of The Review:**

This paper proposes a novel loss function with underlying ideas of local partical density/interactions and mesh-free sampling strategy. It is a good paper with clearly written and organized context. The limitation of this paper could be due to its limited novelty and its general applicability.

---

> ### Author Response · Authors · 2022-11-18
> **Response to the Review**
>
> We thank the reviewer for reviewing our work, and also for recognizing the novelty of our proposed approach.
>
> >There lacks novel proposal of network architecture, which might limit the novelty of this paper. Here, it is worthwhile noting that the novel definition of loss function does qualify as a good contribution. However, the architecture of neural network is not novel.
>
>  We indeed do not introduce a new architecture, as the limitations of PINNs in settings of sparse signals and high dimensions are independent of the network architecture and pose a more fundamental problem.
>
>
> >Author discussed the applicability of the propose methods. However, the limitation of applicability seems to be obvious as the authors stated. But, it is rather an unfamiliar area for me.
>
>  We consider PINNs in settings where we start with a process that is measured (e.g. the density and velocity of birds) and that we want to model.
> Furthermore, we assume additional PDEs are available and can be used as prior knowledge for regularizing the network (e.g. the continuity equation).
> If in this given physical problem there is a non-negative integrable quantity that can be interpreted as a density, and this density is relevant to the dynamics of the PDE, then our method is applicable.
>
> In practice, this is the case for PINNs because measurements are obtained from the physical process itself, and the PDEs are solved in that particular context.
> We want to clarify that this density might also be implicit and not directly measured. Many PDEs start out with e.g. the continuity equation for the conservation of mass where a density is obviously present. By then introducing additional relations between the velocity and density, one can simplify the equation and arrive at a PDE that does not explicitly include the density anymore.
>
> For a discussion regarding the applicability, we would like to refer the reviewer to our top-level comment.

---

### Official Review · Reviewer_cZb3 · 2022-11-03

**Confidence:** 2
**Clarity, Quality, Novelty And Reproducibility:** The paper does not present clearly. I…
**Correctness:** 2
**Technical Novelty And Significance:** 2
**Empirical Novelty And Significance:** 1
**Recommendation:** 3

**Strength And Weaknesses:**

Weaknesses: It looks that I do not understand what the authors' motivation and what they do to solve. Does mesh-free fluid dynamics really work?

**Summary Of The Paper:**

The authors claim to propose to sample directly from the distribution over the particle positions, eliminating  boundaries while adaptively
focusing on the most relevant regions. It looks higher sample efficiency and improved performance of PINNs.

**Summary Of The Review:**

The author need to compare their work to compare the classical work about numerical solution of PDE.

---

> ### Author Response · Authors · 2022-11-18
> **Response to the Review**
>
> We thank the reviewer for the comments and hope we are able to clarify our motivations and proposed solution.
>
> >It looks that I do not understand what the authors' motivation and what they do to solve.
>
>  Our main motivation lies in the existing limitations of PINNs in settings with sparse signals and possibly unbounded domains.
> In general, PINNs require collocation points that explore the whole domain, so that the PDE may be successfully enforced, analogously to the meshes required for finite element methods.
> In PINNs this is commonly done by uniformly sampling from the domain.
>
>
> This causes problems in two settings that we consider:
> 1. If we have an unbounded space domain, fully exploring the domain is ill-posed as we simply can not uniformly sample from e.g. $\Omega=\mathbb{R}^3$.
> 2.  Even if we have a bounded space domain, uniformly sampling in a high-dimensional space quickly becomes infeasible for extensively covering the whole domain. Instead, one should focus on regions of high relevance for regularizing the network.
>
>  These settings do appear in practical applications, with our example being the modeling of migrating birds based on weather radar measurements.
>  In settings where a non-negative integrable scalar quantity is measured and of interest, we propose to sample from the underlying particle distribution.
>
>  We empirically show through multiple experiments that in such settings we require fewer collocation points to achieve a similar test error compared to alternative sampling methods.
>
> > Does mesh-free fluid dynamics really work?
>
> Yes, this is what our paper proposes.
> With respect to initial value problems approached by classical methods, we refer to the extensive literature on mesh-free fluid dynamics, with an overview provided in [1].
>
> ### References
> [1] Li, Shaofan, and Wing Kam Liu. "Meshfree and particle methods and their applications." Appl. Mech. Rev. 55.1 (2002): 1-34.

---

### Official Review · Reviewer_jfSc · 2022-11-03

**Confidence:** 3
**Correctness:** 3
**Technical Novelty And Significance:** 2
**Empirical Novelty And Significance:** 2
**Recommendation:** 6

**Clarity, Quality, Novelty And Reproducibility:**

The paper is clearly written. It seems that some related work may be missing, for instance - https://www.researchgate.net/publication/359932814_Monte_Carlo_PINNs_deep_learning_approach_for_forward_and_inverse_problems_involving_high_dimensional_fractional_partial_differential_equations
seems to have the same line of inquiry, while they focus on fractional PDEs. It is possible that other works exist employing MCMC in similar ways.

**Strength And Weaknesses:**

Strengths:

1) The paper is well written, clearly explains the objective and modifications made to existing methods.
2) The presented method is straightforward to implement, and seems to outperform other methods in similar setups.


Weaknesses/Questions:

1) The suggested method seems to apply only to micro to macro phenomena, however PDEs are ubiquitous in other scenarios, making this solution very restrictive (albeit interesting).
2) "We argue that due to the locality of particle interactions, the regions with higher density are more relevant for regularizing the network." This statement is made as motivation for the class of PDEs under study, but it is not proven.
3) How dependent is the adaptation process on the initial choice of distribution?
4) While the authors compare their results against other PINN based solvers, it is unclear how the performance stands against other non-PINN methods.

**Summary Of The Paper:**

The authors consider a modification of the PINN construction for solving certain classes of partial differential equations. Namely, they focus on PDEs formulated as a macroscopic description of an underlying microscopic physical process, such as diffusion and advection. They claim that the soft constraint on the PINN loss which informs the network of the original PDE is usually taken as uniformly discretized, which is a poor approximation when the true underlying solution distribution of collocation points is not uniformly spread. They suggest replacing the uniform sampling with a Monte Carlo sampling method, in which the distribution itself is updated during training. They claim that this method resolves the issues of unbalanced density solutions and the need to define a bounding region for the PDE loss.
The authors show improved performance over several other collocation point sampling methods, such as uniform, Residual adaptive refinement and some variants, as well as importance sampling.



**Summary Of The Review:**

A well written paper with clear exposition and empirical results, albeit in a limited class of problems.

---

> ### Author Response · Authors · 2022-11-18
> **Response to the Review**
>
>
> We thank the reviewer for the feedback and for mentioning possibly missed literature.
>
> As questions 1. and 2. are related to the applicability of the proposed approach, we refer to our top-level comment where we try to clarify the concerns.
>
> > How dependent is the adaptation process on the initial choice of distribution?
>
>  We presume that this question refers to the choice of the initial sampling distribution, before the network has been trained on data.
> As we stated in the Appendix A.1, "we found that all of these methods work well in practice".
> That is, the initial choice did not play a large role.
> One could also skip it, and only start sampling collocation points once the network has been trained on data for at least one epoch.
>
>
> > While the authors compare their results against other PINN based solvers, it is unclear how the performance stands against other non-PINN methods.
>
>  We provide an improvement to the current state of PINNs, which is why it is the focus of our comparisons.
> Classical methods within the area of data assimilation would require introducing a mesh, which quickly becomes infeasible and very involved for the settings we consider (up to 3+1 dimensional).
>
>
> > It seems that some related work may be missing, for instance [...]
>
> Although the related uses MCMC for PINNS, the referenced paper considers a completely different application of MCMC for a very specific subset of PDEs.
> Specifically, they use MCMC for calculating the derivatives in fractional PDEs, as they can not be calculated using automatic differentation.
> With pdPINNs we still obtain the derivatives by using automatic differentation.
> MCMC for us is just a vehicle for sampling the positions where we evaluate the PDE loss.
> Furthermore, the mentioned work is at the time of the review still an unpublished preprint.
>
> > It is possible that other works exist employing MCMC in similar ways.
>
> If this is the case, we are not aware of it, and would kindly ask the reviewer to point out such works.
>
>  We further emphasize that our contribution does not lie in the use of MCMC, but rather in the proposition to sample from the density.
> The method to use for that can be freely chosen, may that be Variational Inference, MCMC, or some other approach.

---

### Official Review · Reviewer_N5cW · 2022-11-04

**Confidence:** 4
**Correctness:** 3
**Technical Novelty And Significance:** 2
**Empirical Novelty And Significance:** 2
**Recommendation:** 5

**Clarity, Quality, Novelty And Reproducibility:**

The clarity of the exposition is exemplary, with a very strong relation back to prior work in the field, and placing the proposed extension in the right, nuanced context. With previous approaches to the refinement problem in PINNs already utilizing adaptive refinement, and the previous state-of-the-art residual adaptive refinement utilizing Monte Carlo integration at its core, one is lead to question just how close to the proposed approach a particle-based sampler within residual adaptive refinement would be.

In some of the arguments there is some imprecision such as the argument that only a number of systems can be viewed as particle-based systems. In theory every physical systems can be viewed as an artificial particle-based system, but that viewpoint may not always be conducive to the search for a solution, or be able to compute desired quantities of interest.

There furthermore exist too expansive claims, which are not supported by the provided experimental evaluation. To cite "As we have demonstrated applicability to a wide spectrum of PDEs..", I believe this claim is not supported as while the experiments are chosen from different fields, they still a fraction of the potential PDE-dynamics, and as already pointed out above there exist a number of PDEs in PINN-literature which are significantly more difficult.

**Strength And Weaknesses:**

Strengths:
- Strong embedding of the proposed approach into the current state of the literature with illustrative parallels being drawn to previous PINN approaches.
- Well-motivated approach with a clear physical intuition behind it.

Weaknesses:
- As the approach is particle-based I would have expected an ablation analysis/comparison to particle-based inference approaches such as Sequential Monte-Carlo. The authors only consider inverse transform sampling, Metropolis Hastings, and Hamiltonian Monte-Carlo.
- Limited set of experimental evaluations with fairly simple physical systems being considered. Would the proposed approach also work on the more difficult Korteweg-de-Vries equations? Would it work with Burgers equations, and how would it compare to the residual adaptive refinement approach of Lu et al. there? Would it scale to even more complex settings, such as the ones explored in XPINNs?

**Summary Of The Paper:**

The authors propose an extension to the physics-informed neural network approach, which utilizes a particle-based approach to more efficiently sample in the data space, and capture boundary conditions more accurately with less required samples as compared to traditional collocation points. The core contributions of the paper is the injection of microscopic fluid dynamics views rooted in smoothed particle hydrodynamics, into the mesh-free PINN approach to achieve more optimal adaptive sampling.

**Summary Of The Review:**

The authors propose an extension to the PINN training approach rooted in particle-based adaptive refinement, utilizing Monte-Carlo sampling to capture the different regions, and boundaries of the problem more accurately. While well-motivated, and well-derived the approach is only a slight extension beyond the previous residual adaptive refinement approach of Lu et al., which furthermore also already utilized a Monte-Carlo integration step. The experiments, while illustrative, could take more difficult PDE-systems into consideration to showcase the true capability and limitations of the proposed approach.

---

> ### Author Response · Authors · 2022-11-18
> **Response to the Review**
>
> We thank the reviewer for the interesting comments and for the suggestions.
> In the following, we will try to clarify some concerns raised in the review.
>
> >As the approach is particle-based I would have expected an ablation analysis/comparison to particle-based inference approaches such as Sequential Monte-Carlo.
>
>
> We agree that Sequential Monte-Carlo is in general an interesting suggestion as a sampler, especially due to its relation to particles.
> However, there are two major issues, which lead us to not make use of sequential samplers.
>
> 1. PINNs usually include data that is not necessarily measured at the initial time step, as for example in our Continuity Equation experiments. In these settings, the network first learns a density where we also have data, and the density is thus not yet uniformly distributed in time. As Sequential Monte-Carlo is by construction sequential in time, it is not straightforward how to apply it to such a setting. Nonetheless, for initial value problems, it might be a sensible approach.
> 2.  Sequential Monte Carlo would require a discretization of time. A-priori it's not clear how to choose a sufficiently small step size while also keeping enough samples to explore the space. By directly sampling according to the density distribution over both space and time, we leave this to the sampler.
>
>
> > Would the proposed approach also work on the more difficult Korteweg-de-Vries equations? Would it work with Burgers equations and how would it compare to the residual adaptive refinement approach of Lu et al. there?
>
>
> This work concerns PINNs in settings where we have measurements of a physical process and additional PDEs that provide us prior knowledge.
> Whenever in this physical process there is a notion of density, and this density is relevant for the dynamics of the process, pdPINNs are applicable.
> However, just providing an equation without context is usually insufficient for answering the question, as for example the Burgers equation might be derived from multiple different settings.
> For a discussion of this, we refer to our top-level comment, where we try to provide a more concrete explanation of the applicability of pdPINNs, with the example of the 1d Burgers equation.
>
> Furthermore, our approach shines when the measured signal is heavily concentrated on a (small) subset of the domain, especially in large dimensions.
> If there is roughly equal density in the whole domain, sampling from the density will not be much different from uniform sampling.
> Again, whether this is the case depends strongly on the actual physical problem at hand, not just the dynamics given by the equation.
>
> >Would it scale to even more complex settings, such as the ones explored in XPINNs?
>
>  XPINNs suggest an approach for a domain decomposition that allows training separate PINNs within each sub-domain.
> In each sub-domain, one still has to sample collocation points (in the XPINN paper referred to as residual points).
> Our proposed pdPINNs provide a way to sample these points.
> In that sense, the two methods do not really compete with each other.
> On the contrary, they should nicely complement each other.
> It should be straight forward to add our sampling procedure as the used sampling mechanism in XPINNs.
> XPINNs, however, do not explore any specific more complex setting.
> In the paper [1] they are evaluated on a 1-dimensional initial value problem for the viscous Burgers equation.
>
>
>
> > [..] while the experiments are chosen from different fields, they still a fraction of the potential PDE-dynamics, and as already pointed out above there exist a number of PDEs in PINN-literature which are significantly more difficult.
>
>  Our contribution focuses on the difficulties arising due to spatially sparse signals, which are amplified in high dimensions.
> The settings considered in the PINN literature often restrict themselves to one-dimensional problems (+ Time), or sometimes 2 dimensional (+ Time) settings, which are with respect to that much simpler.
> In these settings, the challenges of sparse signals can still be overcome by simply using more collocation points.
> In our experiments, we consider settings with up to 3 + 1 dimensions, where extensively exploring the input domain based is simply infeasible.
>
> ### References
>
> [1] Jagtap, Ameya D., and George E. Karniadakis. "Extended Physics-informed Neural Networks (XPINNs): A Generalized Space-Time Domain Decomposition based Deep Learning Framework for Nonlinear Partial Differential Equations." AAAI Spring Symposium: MLPS. 2021.

---

### Author Response · Authors · 2022-11-18
**Response to All Reviewers: Common Questions**

We are thankful to all the reviewers for providing their feedback and suggestions for our submitted paper.

We noticed two main concerns that are shared between the reviewers:
1. Questions about the applicability of pdPINNs.
2. Concerns about the use of MCMC for sampling collocation points.

We expect to be able to clarify some of these concerns.

## Applicability of pdPINNs

There have been some questions regarding the extent of the applicability of pdPINNs, and whether they can also be used for some very specific PDEs.
In general, we consider PINNs to be relevant in settings where we model a physical process based on available measurements.
In such settings, pdPINNs are applicable whenever the measured physical quantity is non-negative and integrable and locally relevant to the dynamics of the PDE.
Whether this is the case should be obvious from the physical application setting, and can not be decided based solely on the PDE.

For example the 1d Burgers equation arises from a range of different settings [1].
One instance is a traffic flow problem where we would for example measure the density and possibly also the velocity of cars.
For deriving the dynamics of this problem, one starts from the continuity equation for the conservation of mass, i.e. of cars.
By now introducing an equation relating density and velocity, one can express the two equations just in terms of the velocity and arrives at the 1d inviscid Burgers equation [2].
Note that in this case the density is not explicitely present in the PDE, but we are still able to use pdPINNs.

In summary, pdPINNs are applicable in settings where we measure a non-negative scalar quantity, such as a (unnormalized) particle or charge density, that satisfies a conservation equation.
Furthermore, additional measurements such as the velocity or pressure might be available.
More complex PDE systems describing the dynamics of the measured quantity can then be derived by introducing additional equations, which for example relate velocity and density.


## Concerns about the use of MCMC for sampling collocation points.
Multiple concerns have been raised whether MCMC is the ideal choice for the sampler, and want to clarify that our contribution does not lie in proposing the use of MCMC with PINNs.
Instead, we propose to sample collocation points from the particle distribution, which is only available in its unnormalized form via a neural network. In this framework we regard MCMC as one way to sample from an unnormalized distribution. In particular, we chose MCMC due to its theoretical guarantees that the Markov Chain eventually converges to the target distribution, which is ideal for showcasing the benefits of sampling from another density. However, alternative sampling methods can of course be used. For this reason, we additionally included inverse transform sampling, which is extremely fast but will struggle with sampling from the target distribution in high-dimensions. Other sampling procedures such as Sequential Monte-Carlo and Variational Inference based methods still lie within our framework. However, even when using MCMC, we show in the Appendix Figure A.10 that Metropolis Hastings does not require much more runtime than uniform sampling, and Inverse Transform sampling is as fast as Uniform sampling while outperforming it in every experiment. Overall, the user has complete freedom in choosing the more appropriate sampler according to the problem under study, which we consider as one of the strengths of our approach.


### References

[1] Bonkile, Mayur P., et al. "A systematic literature review of Burgers’ equation with recent advances." Pramana 90.6 (2018): 1-21.

[2] Jüngel, Ansgar. "Modeling and Numerical Approximation of Traffic Flow Problems." Lecture Notes (preliminary version) (2002).

---

### Decision · Program_Chairs · 2023-01-20

**Decision:**

Reject

**Justification For Why Not Higher Score:**

Limited novelty and questions about the applicability to different types of differential equations.

**Justification For Why Not Lower Score:**

N/A

**Metareview: Summary, Strengths And Weaknesses:**

The paper proposes an extension of physics-informed neural networks (PINNs) for solving specific classes of PDEs. The vanilla PINN method makes use of a uniform sampling for enforcing physical constraints as provided by the PDEs. The authors point out that this is inefficient when the true solution is not uniformly distributed and restrictive since it cannot handle unbounded domains. They propose a MCMC adaptive sampling method motivated by a microscopic viewpoint of fluid dynamics formalized by the so called “molecular distribution function” that improves upon uniform sampling.  This is evaluated on three applications.

The paper is well motivated and clear. The reviewers however consider that this work is only a slight extension of existing approaches since adaptive sampling have already been proposed for this family of models. They also question the applicability of the method which seems limited to some subclasses of equations and consider that the experiments are insufficient to support the claims. The authors responses did not change the reviewers’ opinion.